# Cross-movie prediction of individualized functional topography

Guo Jiahui[1], Ma Feilong[1], Samuel A Nastase[2], James V Haxby[1], M Ida Gobbini[3,4]*

[1]Center for Cognitive Neuroscience, Dartmouth College, Hanover, United States; [2]Princeton Neuroscience Institute, Princeton University, Princeton, United States; [3]Department of Medical and Surgical Sciences (DIMEC), University of Bologna, Bologna, Italy; [4]IRCCS, Istituto delle Scienze Neurologiche di Bologna, Bologna, Italy

**Abstract** Participant-specific, functionally defined brain areas are usually mapped with functional localizers and estimated by making contrasts between responses to single categories of input. Naturalistic stimuli engage multiple brain systems in parallel, provide more ecologically plausible estimates of real-world statistics, and are friendly to special populations. The current study shows that cortical functional topographies in individual participants can be estimated with high fidelity from naturalistic stimuli. Importantly, we demonstrate that robust, individualized estimates can be obtained even when participants watched different movies, were scanned with different parameters/ scanners, and were sampled from different institutes across the world. Our results create a foundation for future studies that allow researchers to estimate a broad range of functional topographies based on naturalistic movies and a normative database, making it possible to integrate high-level cognitive functions across datasets from laboratories worldwide.

## Editor's evaluation

This valuable study presents a tool for hyperaligning functional brain topography between individuals, which is based on fMRI connectivity data gathered when participants watched different movies. The tool is validated through strong correlations between functional topographic maps generated from a participant's own localizer data and those derived from other participants' data based on this hyperalignment, even when the training and target participants were drawn from different datasets. The study will potentially be of interest to researchers working with a wide range of fMRI datasets.

*For correspondence: mariaida.gobbini@unibo.it

Competing interest: The authors declare that no competing interests exist.

## Introduction

Category-selective functional topographies are a prominent and consistent feature of lateral occipital, ventral temporal, and lateral temporal visual cortices (*Downing et al., 2001*; *Epstein et al., 1999*; *Grill-Spector and Weiner, 2014*; *Kanwisher et al., 1997*). Category-selective topographies are mostly similar across individuals but are idiosyncratic in terms of their precise conformation and location (*Zhen et al., 2015*; *Zhen et al., 2017*). Because of these idiosyncrasies, category-selective topographies and areas are typically mapped in each individual using a functional localizer fMRI scan (*Fedorenko et al., 2010*; *Saxe et al., 2006*). Functional localizers map individualized topographies with simple contrasts between responses to different categories, such as contrasting responses to faces versus objects to localize face-selective areas.

We reported an alternative approach to map category-selective topographies using fMRI data collected while participants view a naturalistic movie (*Guntupalli et al., 2016*; *Haxby et al., 2011*; *Jiahui et al., 2020*). With this approach, movie-viewing and functional localizer data are collected in a normative sample, and new participants need only be scanned during movie viewing. Movie data

are used to calculate transformation matrices using hyperalignment (*Guntupalli et al., 2016*; *Haxby et al., 2011*; *Jiahui et al., 2020*; *Feilong et al., 2018*; *Feilong et al., 2021*; *Feilong et al., 2021*; *Guntupalli et al., 2018*) that afford projecting the localizer data from the normative sample into the idiosyncratic cortical topography of new participants. Using this hyperalignment procedure, we can estimate the idiosyncratic details of individual topographies with high fidelity based on localizer data from the normative sample. Unlike functional localizers, naturalistic stimuli (e.g., movies) evoke a rich variety of brain states and engage multiple brain systems in parallel. This makes it possible to efficiently map multiple functional topographies using data from a single movie and avoid the time and cost of running multiple localizers. Compared to controlled localizers, movies better simulate real-world cognition and better engage participants' attention (*Vanderwal et al., 2015*; *Vanderwal et al., 2017*; *Vanderwal et al., 2019*), contributing to more ecologically valid and higher-quality maps. In addition, movies are more friendly and engaging for special populations, such as young children.

In previous work, we used response hyperalignment (RHA) to predict functional topographies in new participants. RHA requires that all participants watch the same movie to obtain time-locked responses to the same stimuli. It is often important, however, to tailor the movie to meet the specific needs of participants in different experiments. For example, participants from different countries may prefer movies that reflect their diverse backgrounds and are in their native languages (*Hanke et al., 2016*; *Sengupta et al., 2016*); movies for infants and young children are differently structured from those for adults (*Vanderwal et al., 2015*). Thus, it is unrealistic to limit all participants from diverse populations and backgrounds to watch the same movie. Additionally, experimenters may need to shorten or edit the stimuli to fit their data collection schedule. Finally, participants are often scanned with different parameters from one experiment to another, at different institutes across the world, and with different scanner models. Due to these factors, it is impractical to expect two laboratories to acquire the same movie scans across individuals.

Here, we test whether connectivity hyperalignment (CHA) (*Guntupalli et al., 2018*) can be used to map category-selective functional topographies. CHA, in contrast to RHA, affords calculation of transformation matrices using stimuli that are not the same for normative and index participants. We analyzed four different datasets collected with three different movies, three different scanners, and two different types of functional localizers that used dynamic or static stimuli. We first demonstrated that CHA based on participants' connectomes that were calculated using their responses to movies was able to generate high-fidelity maps of category-selective topographies within datasets that were equivalent to maps estimated using RHA. Then, critically, we showed that cross-dataset predictions that used connectomes calculated from different movies for the normative and index brains were as good as those from participants in the same dataset. This means that different laboratories can use different movies to derive functional topographies from a normative sample.

In summary, we demonstrate that a target participant's individualized category-selective topography can be accurately estimated using CHA, regardless of whether different movies are used to calculate the connectome and regardless of other data collection parameters. Movies engage multiple cognitive domains in parallel, such as visual perception, audition, language comprehension, theory of mind, and social interaction. In addition to estimating different functional topographies from a single movie, our approach allows us to estimate topographies from different movies. We provide a novel alternative for future data collection that can save time and money using rich and efficient movie scans.

## Results
### High-fidelity prediction with CHA
We predicted category-selective topographies by projecting other participants' functional localizer data into each participant's native cortical topography using a new, enhanced CHA algorithm. For each participant, we calculated transformation matrices based on functional connectivity estimated during movie viewing in an iterative way (see Materials and methods). These transformation matrices resample fMRI data from others' brains into a given participant's cortex. We then projected the functional localizer data for all other participants into the given participant's native cortical space and calculated independent functional contrasts based on that participant's own localizer data and based on other participants' localizer data projected into that participant's cortex. We also estimated

functional topographies by projecting others' localizer data into that participant's cortex based on high-performing surface-based anatomical alignment as a control analysis. We calculated the correlations between topographies based on participants' own localizer contrasts and on other participants' data. Because the localizer task comprises several scanning runs, we calculated the reliability of the localizer across runs with Cronbach's alpha to provide an estimate of the noise ceiling for these correlations. We repeated this procedure for all participants.

We tested the estimation of visual category-selective functional topographies (faces, bodies, scenes, and objects) in four different datasets using three different movies, localizers with static or dynamic stimuli, different scanning sequence parameters, and three different scanner models (see Materials and methods).

Category-selective topographies estimated with CHA recovered the idiosyncrasies of individuals' topographies, capturing fine details of the individual-specific configuration and extent. By contrast, topographies estimated with anatomical alignment generated highly blurred maps that were essentially the same for all participants, losing individual-specific idiosyncratic features (*Figure 1A*).

The superior performance of CHA-based estimation over anatomical-alignment-based estimation was consistent across participants, visual stimulus categories, and datasets. In all four category-selective topographies and in all four datasets, correlations between estimations based on hyperalignment and their own localizer data were significantly higher than the correlations between estimations based on anatomical alignment and each participant's own localizer (Fisher z-transformed, $p < 0.001$, Bonferroni corrected). We compared these correlations between topographies estimated from a participant's own localizer data and those from other participants' data to the reliability of the localizer, calculated with Cronbach's alpha. Predictions made with hyperalignment were close to and sometimes even exceeded the reliability values (*Figure 1B*), which indicate that the predicted category-selective topographies from other participants' data using hyperalignment were as precise and sometimes even better than the topographies estimated with their own localizer data.

Estimates using CHA to calculate transformation matrices were also equivalent to estimates using RHA (*Figure 1D*). RHA, however, requires that all subjects watch the same movie, whereas CHA can use connectivity matrices derived from responses to different movies, potentially making our new approach more flexible. Next we tested the validity of estimating topographies using transformation matrices that were based on functional connectivities calculated from responses to different movies for the test participant and other participants.

## CHA enables cross-movie predictions

Experimental design considerations and constraints can make using the same stimulus across all studies and participants inadvisable, and datasets are often collected under diverse conditions. Here, we aim to test whether connectivity-based hyperalignment can predict category-selective topographies in new individuals even if their connectomes are estimated from data collected while they watched a different movie. Using this method, participants across datasets without matched time-locked functional series can benefit from those who have functional localizer data but were scanned with different naturalistic stimuli.

We estimated category-selective topographies for each participant in each dataset from participants in the other dataset that used the same type of localizer (dynamic or static) by calculating transformation matrices based on functional connectivities measured while watching different movies. We also estimated topographies based on anatomical alignment. The cross-movie predictions using CHA outperformed predictions based on anatomical alignment and were nearly as precise as within-movie predictions (*Figure 2A*). The superior performance was consistent across datasets and categories ($p < 0.001$ for all comparisons, *Figure 2B*) and in all individual participants (*Figure 2—figure supplement 2*). Similarly, accuracies of these predictions matched and sometimes even exceeded the reliability measures of their own localizer runs (*Figure 2B*).

Cross-movie predictions of cortical topographies based on different localizer types (static to dynamic or dynamic to static) produced lower correlations than did cross-movie predictions based on the same localizer type (*Figure 2—figure supplement 1*), consistent with previous reports showing significant differences between topographies estimated by static and dynamic localizers, especially in superior temporal and frontal cortices (*Fox et al., 2009*; *Pitcher et al., 2011*).

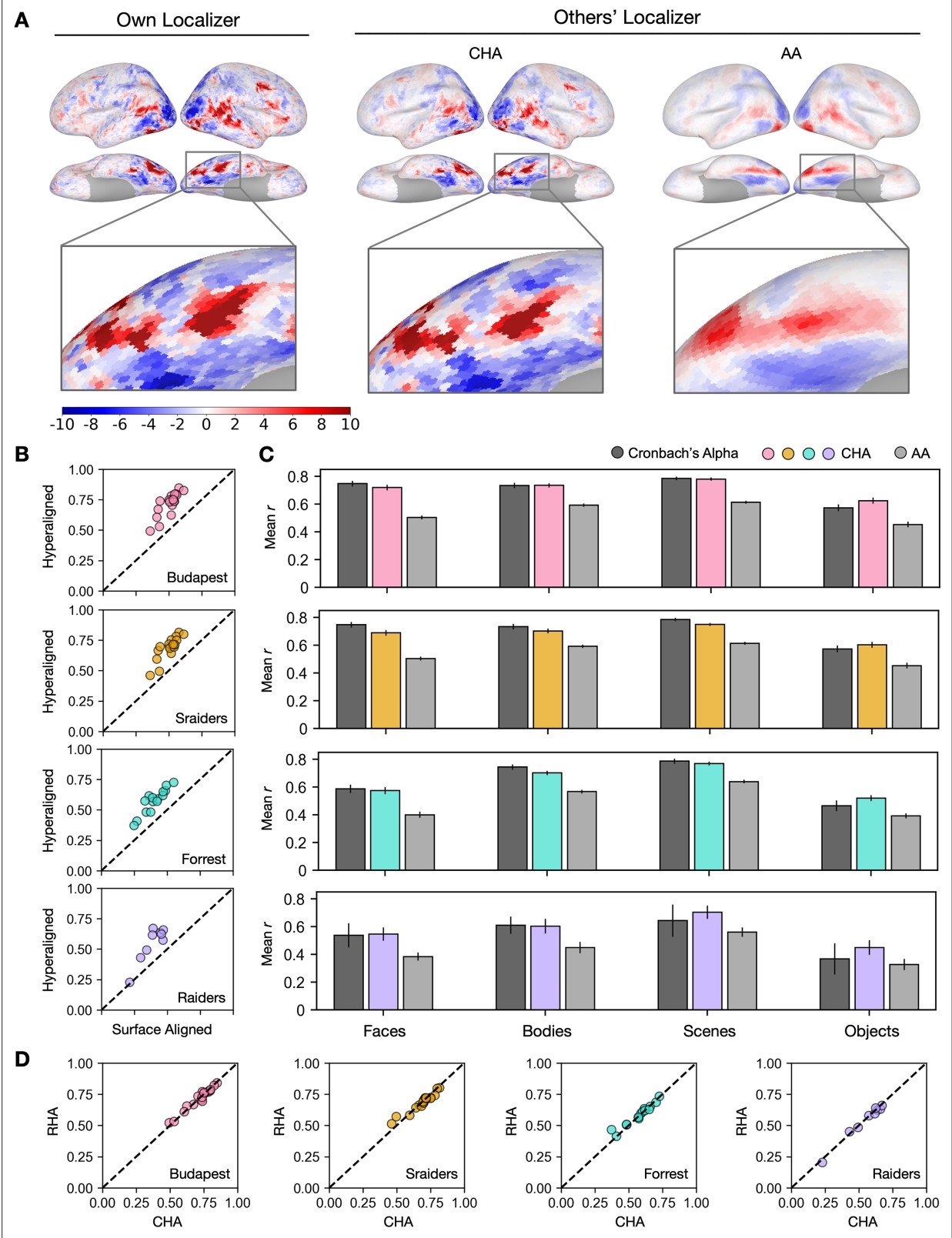

**Figure 1.** Predicting individual category-selective topographies using connectivity hyperalignment (CHA). (**A**) Face-selective topographies (faces-vs-all) and zoomed-in views of an example participant estimated from this participant's own localizer (Own Localizer), and other participants' localizers using CHA, and surface anatomical alignment (AA). (**B**) Scatter plots display the Pearson correlation coefficients between estimated face-selective topographies based on own localizer data and other participants' localizer data in individual participants in four different datasets. The y-axis

*Figure 1 continued on next page*

*Figure 1 continued*

corresponds to correlations between each target participant's own localizer-based face-selective topographies and face-selective topographies estimated from other participants using CHA. The x-axis corresponds to correlations between each target participant's own localizer-based face-selective topographies and face-selective topographies estimated from other participants with surface-based anatomical alignment. (**C**) Bar plots show the mean correlations across participants in four datasets (Budapest & Sraiders: *n* = 20; Forrest: *n* = 15; Raiders: *n* = 9. Same sample sizes in other figures for each dataset unless noted.) and for all four category-selective topographies. Black bars stand for the mean Cronbach's alphas across participants. Error bars indicate ±1 standard error of the mean. Category topographies were defined based on contrasts between the target category and all other categories. (**D**) Scatter plots of Pearson correlation coefficients using CHA and response hyperalignment (RHA) for individual participants within four different datasets for the face-selective topography. Values on the y-axis stand for correlations between each target participant's own localizer-based topographies and topographies estimated from other participants in the same dataset using RHA. Values on the x-axis stand for correlations between each target participant's own localizer-based topographies and topographies estimated from other participants in the same dataset using CHA.

The online version of this article includes the following figure supplement(s) for figure 1:

**Figure supplement 1.** Schematic data analysis procedures.

To demonstrate how hyperalignment increased prediction performance for individual participants from a different dataset, we plotted topographies estimated using hyperalignment and anatomical alignment, as well as from their own localizer runs (*Figure 3*, *Figure 3—figure supplement 1* and *Figure 3—figure supplement 2*). Topographies between datasets recovered similar idiosyncratic features as the topographies predicted within datasets.

To further examine the topographies predicted using different datasets and compare the prediction performances to reliability measures, we calculated local correlations between maps estimated from each participant's own localizer runs and those estimated from other participants' runs with a searchlight analysis. We also calculated Cronbach's alpha across localizer runs in each searchlight. Generally, searchlights in the high-level visual areas and with strong category selectivity (e.g., ventral temporal cortex, lateral temporal cortex) showed the highest mean correlation values, which often exceeded 0.8 (*Figure 4*, *Figure 4—figure supplement 1*, *Figure 4—figure supplement 3*, and *Figure 4—figure supplement 10*). The lower mean correlations in other cortices (e.g., sensorimotor cortex) reflect low reliabilities of the localizer runs.

## Discussion

In this study, using four datasets that contain three different movies, two different types of functional localizers, and collected with three different scanners, we showed that individualized category-selective topographies can be estimated with high fidelity using CHA. Unlike RHA, which requires the same 'time-locked' response time series in the normative sample and new participants, CHA affords the calculation of transformation matrices based on responses to completely different movies. By showing that CHA based on participants' connectomes calculated using their responses to different movies generated high-fidelity mappings that were as good as those using RHA with participants in the same dataset, we demonstrated that CHA is able to effectively predict topographies across diverse situations. This study opens new possibilities connecting independent public and in-lab datasets for future data analysis so that researchers can derive multiple topographies at once for each individual with excellent performance based on the naturalistic movie data and the localizer data from another normative dataset. Our results also provide a novel alternative for new data collection to take better advantage of naturalistic stimuli.

We used a new, enhanced CHA in this study that optimized our previous CHA algorithm with iterative steps. In each step, transformation matrices to each index brain were calculated from other participants' brains and the matrices were applied to both the movie and the localizer data. Because using dense connectivity targets (e.g., using all vertices as connectivity targets) with anatomically alignment data often leads to suboptimal alignment across participants (*Hanke et al., 2014*), we started with coarse connectivity targets and gradually increased the number of connectivity targets to form a denser representation of connectivity profiles. The iterations improved the prediction performance step by step, and at the final step (step 6, all vertices were used as connectivity targets) in this analysis, the enhanced CHA generated comparable performance with RHA (*Figure 4—figure supplement 4*). We investigated the influence of naturalistic movie length and the size of the training group on the prediction accuracy of individualized functional topographies. By incrementally increasing both

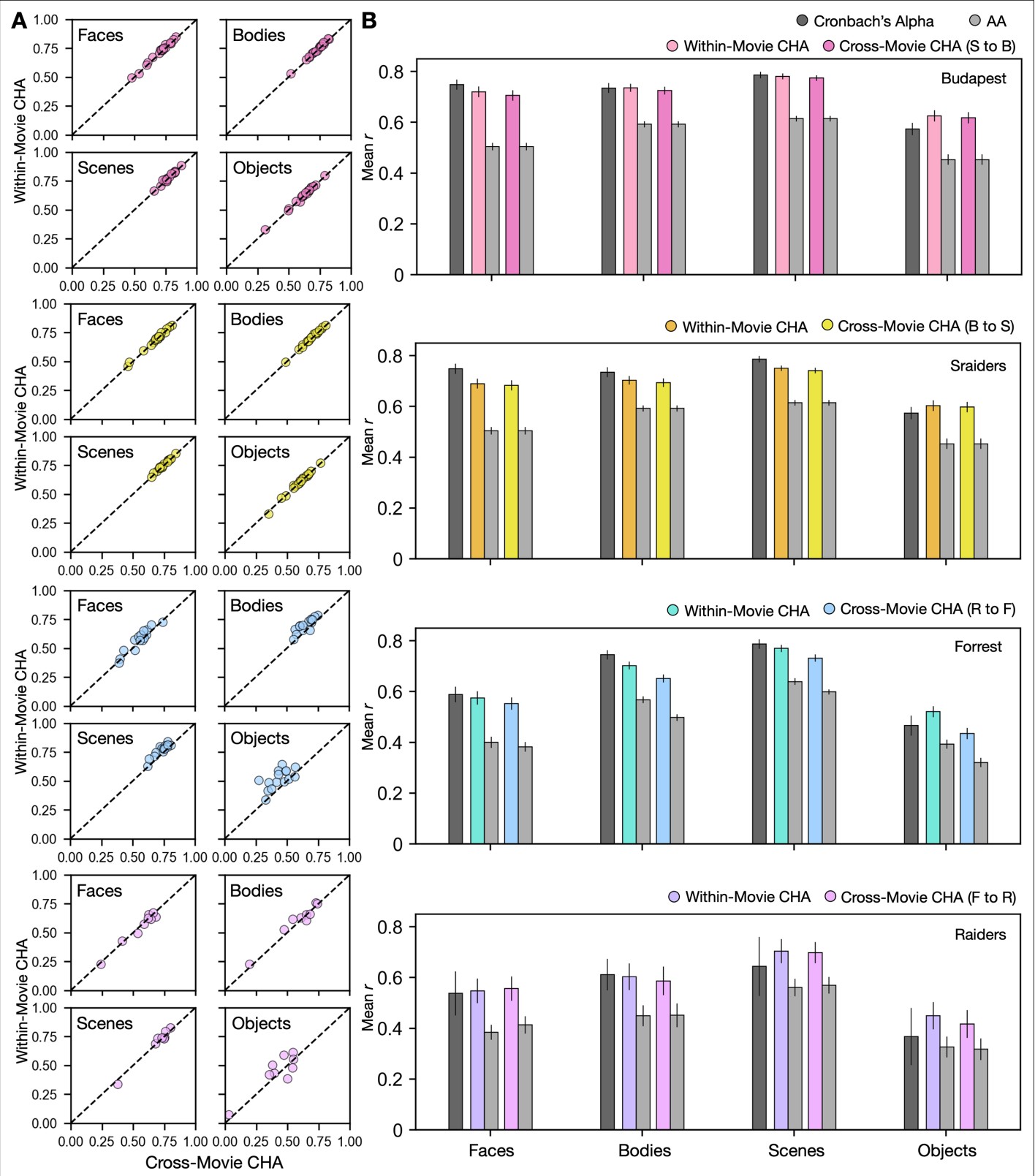

**Figure 2.** Predicting category-selective topographies using connectivity profiles across movies. (**A**) Scatter plots of Pearson correlation coefficients for individual participants in four different datasets and for four categories. Values on the y-axis stand for correlations between each target participant's own localizer-based topographies and topographies estimated from other participants in the same movie using connectivity hyperalignment (CHA). Values on the x-axis stand for correlations between each target participant's own localizer-based topographies and topographies estimated from participants in

*Figure 2 continued on next page*

*Figure 2 continued*

another dataset based on cross-movie CHA. (**B**) Bar plots display the mean Pearson correlation coefficients (r) and Cronbach's alphas across participants in all four datasets for all four categories. Error bars stand for ±1 standard error of the mean. S to B: Sraiders to Budapest, B to S: Budapest to Sraiders, R to F: Raiders to Forrest, F to R: Forrest to Raiders.

The online version of this article includes the following figure supplement(s) for figure 2:

**Figure supplement 1.** Connectivity hyperalignment (CHA) predictions.

**Figure supplement 2.** Prediction performances for each individual participant.

the number of movie runs in the training and target dataset and the participants in the training group in the Budapest and Sraiders dataset, we observed enhanced prediction accuracy (*Figure 4—figure supplement 5*). Notably, even with just one movie run in the training or target dataset, or with a mere five participants in the training group, our prediction performance (Pearson r) ranged from about 0.6 to 0.7. This accuracy significantly outperformed results obtained using surface-based alignment. In addition, this study is based on the new optimized 1-step hyperalignment procedure (*Jiahui et al., 2020*). The classic hyperalignment method (2-step), builds a common information model space at the initial step that is based on all normative group participants, then projects information encoded in idiosyncratic representational spaces to the common model space, and lastly projects the information back to the individual participant's space based on the transpose of the transformation matrices from the former step. Different from the 2-step method, the 1-step method directly projects the data for each normative sample brain to the index participant's space without the intermediate step of building a common information model space. This method requires fewer steps and is free from the accumulation of errors across steps. The 1-step method consistently improved the prediction performances across all conditions and datasets (*Figure 4—figure supplement 6*). This method is particularly useful for estimating information encoded in each individual's brain space. Our original algorithm is designed to apply transformation matrices to the time series of localizer data of training participants before generating contrast maps. To explore whether directly applying these matrices to precalculated contrast maps yields comparable results, we conducted an additional analysis across the four categories. Our findings indicate that the prediction outcomes were indeed quite similar between the two approaches for both the within- and across-datasets predictions (*Figure 4—figure supplement 7*). However, it is worth noting that the improvements observed with enhanced CHA were not as pronounced when applied directly to the contrast maps as opposed to the time series. In our study, we used fine-scale connectomes, noting that some participants are more similar to the target participant in specific searchlights. It is an interesting question whether predictions could be enhanced by exclusively selecting those more similar participants for the target participant. To explore this option, we examined a searchlight in the right ventral temporal cortex that was roughly at the location of the posterior fusiform area using the top and bottom nine participants similar to each target participant measured by their fine-scale connectome similarities in the budapest dataset. Generally, using all or part of the participants for the prediction generated similar results (*Figure 4—figure supplement 8*). Compared to using all the participants, using only the top nine participants who are the most similar to the target participants did not significantly improve the prediction (Tukey test, z=–0.09, p=0.996), but using only the bottom nine participants generated significantly lower prediction accuracies (Tukey test, z=2.492, p=0.034). This suggests a trade-off between the number of participants included in the prediction and the similarity of the participants. Future studies are needed to explore the optimal threshold for the number of participants included for each searchlight to refine the algorithm.

By leveraging transformation matrices obtained from hyperaligning participants based on movie-viewing data, we successfully mapped these relationships to the training participants' localizer data, enabling robust predictions. Prior work employing diffusion-weighted imaging has underscored the link between anatomical connectivity and category selectivity across diverse visual fields (*Osher et al., 2016*; *Saygin et al., 2012*) and has established a notable congruence between structural and functional connectivities (*Hermundstad et al., 2013*). These findings suggest that the unique anatomical connectivity patterns of individuals may serve as a foundational mechanism, contributing to the stable fine-scale functional connectome that underpins our approach. The connectivity-based shared response model (cSRM) proposed by *Nastase et al., 2020*, used connectivity to functionally align individuals similar to the CHA algorithm. While both approaches share overarching goals, they

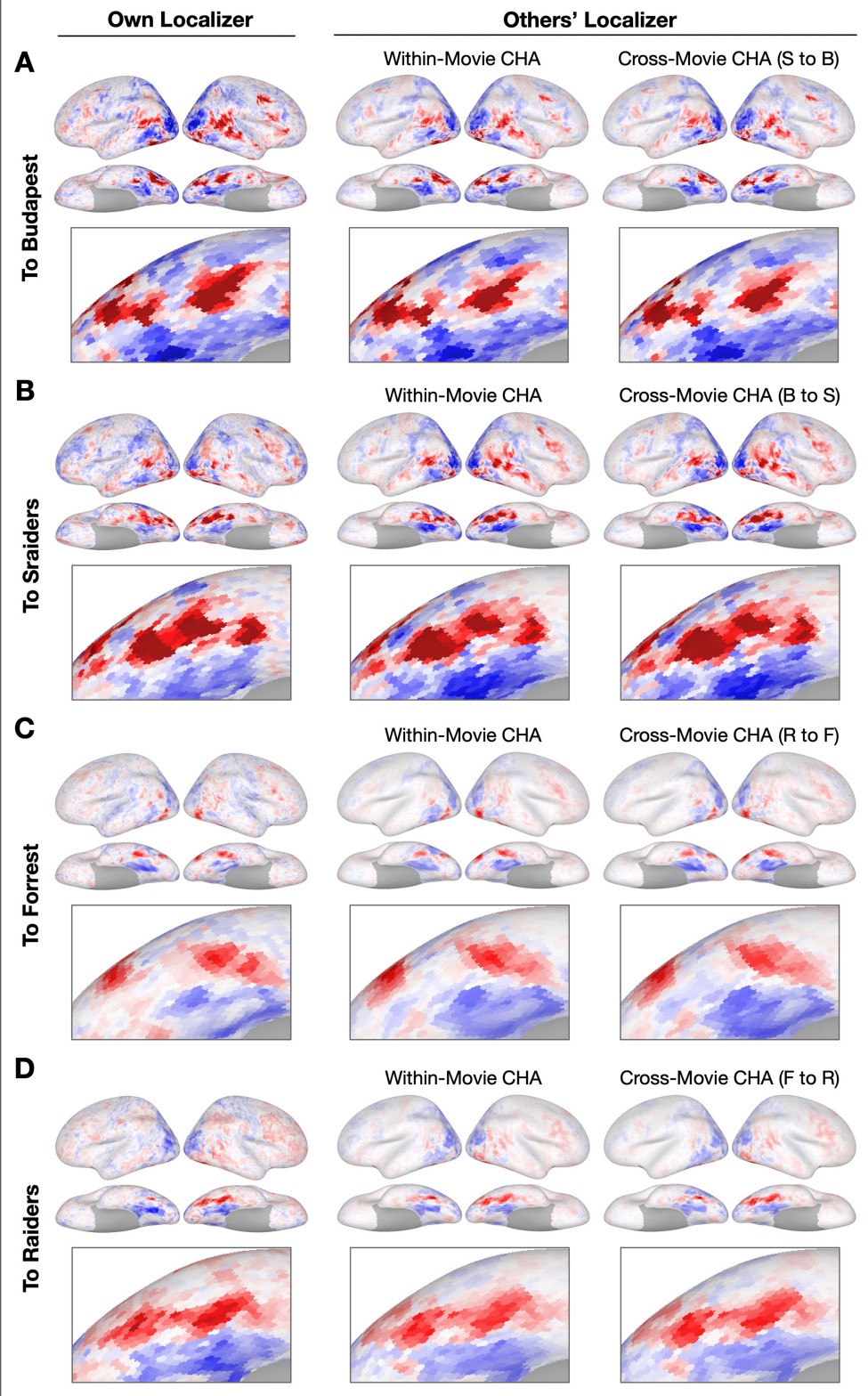

**Figure 3.** Sample contrast maps and enlarged views of the ventral temporal cortex. Contrast maps for face-selective topographies (faces-vs-all) and their zoomed-in views of the ventral temporal cortex were plotted in four sample participants in (**A**) Budapest, (**B**) Sraiders, (**C**) Forrest, and (**D**) Raiders. In all four subplots, in the left-most panel, faces-vs-all maps were plotted on the sample participants' own cortical surfaces. The next two columns display maps estimated from other participants' data. In the right two columns, the first column presents predicted

*Figure 3 continued on next page*

*Figure 3 continued*

face-selective topographies from participants in the same dataset using connectivity hyperalignment (CHA). The next column presents face-selective topographies from participants in another dataset (cross-movie CHA). The zoomed-in panels are displayed accordingly with the whole-brain map. The color bar is the same as that in *Figure 1*. S to B: Sraiders to Budapest, B to S: Budapest to Sraiders, R to F: Raiders to Forrest, F to R: Forrest to Raiders.

The online version of this article includes the following figure supplement(s) for figure 3:

**Figure supplement 1.** Sample contrast maps and enlarged views of the ventral temporal cortex.

**Figure supplement 2.** Sample contrast maps of body-, scene-, and object-selective topographies.

diverge considerably in implementation and application. First and most important, cSRM used inter-subject functional connectivity rather than within-subject functional connectivity to initially estimate the connectome. As a result, cSRM requires participants to have time-locked fMRI time series. Therefore, unlike our algorithm, the cSRM approach does not support cross-content applications and also is not suitable for use with resting-state data. Second, cSRM is implemented based on a predefined cortical parcellation rather than the overlapping, regularly spaced cortical searchlights applied in our method which are not constrained by areal borders. For the application, cSRM has mainly been used to do ROI analysis rather than the estimation of the whole-brain topography that requires broader coverage of the cortex with a searchlight analysis. Third, our method is specifically designed to work in each individual's space, while cSRM decomposes data across subjects into shared and subject-specific transformations, focusing on a communal connectivity space. In summary, although cSRM presents a promising alternative for similar aims, its current implementation precludes it from fulfilling the range of applications for which our method is optimized.

The within-movie and cross-movie CHA predictions generated highly similar topographies (*Figure 3*). This result raises a fascinating question of whether different movie inputs estimate similar fine-grained connectivity profiles in the brain. Previous studies reported that the coarse-grained connectome (based on coarse parcellations) varies across separate cognitive tasks (*Shine et al., 2016*; *Telesford et al., 2016*), and that naturalistic movies yield the most condition-specific functional atlases among other classic cognitive tasks (*Salehi et al., 2020*). In the Budapest and Sraiders datasets, the same group of participants watched the Grand Budapest Hotel and Raiders of the Lost Ark in different sessions in the same 3 T scanner. We built connectivity profiles for each participant separately for the two movies and correlated the two fine-grained connectomes in each searchlight. Results showed that the two fine-grained connectomes based on different movies were very similar in most of the brain regions (r>0.8, *Figure 4—figure supplement 9A, B*). We split each movie into two halves (Run 1–3/Run 4–5 for Budapest; Run 1–2/Run 3–4 for Sraiders) and averaged the connectome similarities across split halves over searchlights and participants. We found that the across-movie connectome similarities for split halves were high (r>0.74), and the within-movie similarities were even higher in both datasets (r>0.85, *Figure 4—figure supplement 9C*). Our analysis showed that although the fine-grained connectome was affected by the input naturalistic stimulus content, it was nonetheless highly stable. This result suggested the brain may undergo shared cognitive processes across different movie free-viewing tasks. It could be because featured movies sample a broad range of real-life statistics, and the rich information elicits overall similar representations and connectivities when the entire time series is considered. Studies comparing movie-viewing and resting-state functional connectivity have shown that both paradigms yield overlapping macroscale cortical organizations (*Samara et al., 2023*), though naturalistic viewing introduces unique modality-specific hierarchical gradients. However, there remains a gap in research comparing the fine-scaled connectomes of naturalistic and resting-state paradigms. *Guntupalli et al., 2018*, revealed a shared fine-scale structure that coexists with the coarse-scale structure, and CHA successfully improved intersubject correlations across a wide variety of tasks. *Feilong et al., 2021*, noted that the fine-scaled connectivity profiles in both resting and task states are highly predictive of general intelligence. This suggests a reliable and biologically relevant fine-scale resting-state connectivity structure among individuals. Therefore, it is plausible that individualized functional topography could be effectively estimated using resting-state functional connectivity, expanding the applicability of our approach. Future studies are needed to explore this direction.

The four datasets in our study included two types of category-selective localizers (dynamic and static). The dynamic localizer used short video clips for each category and the traditional static

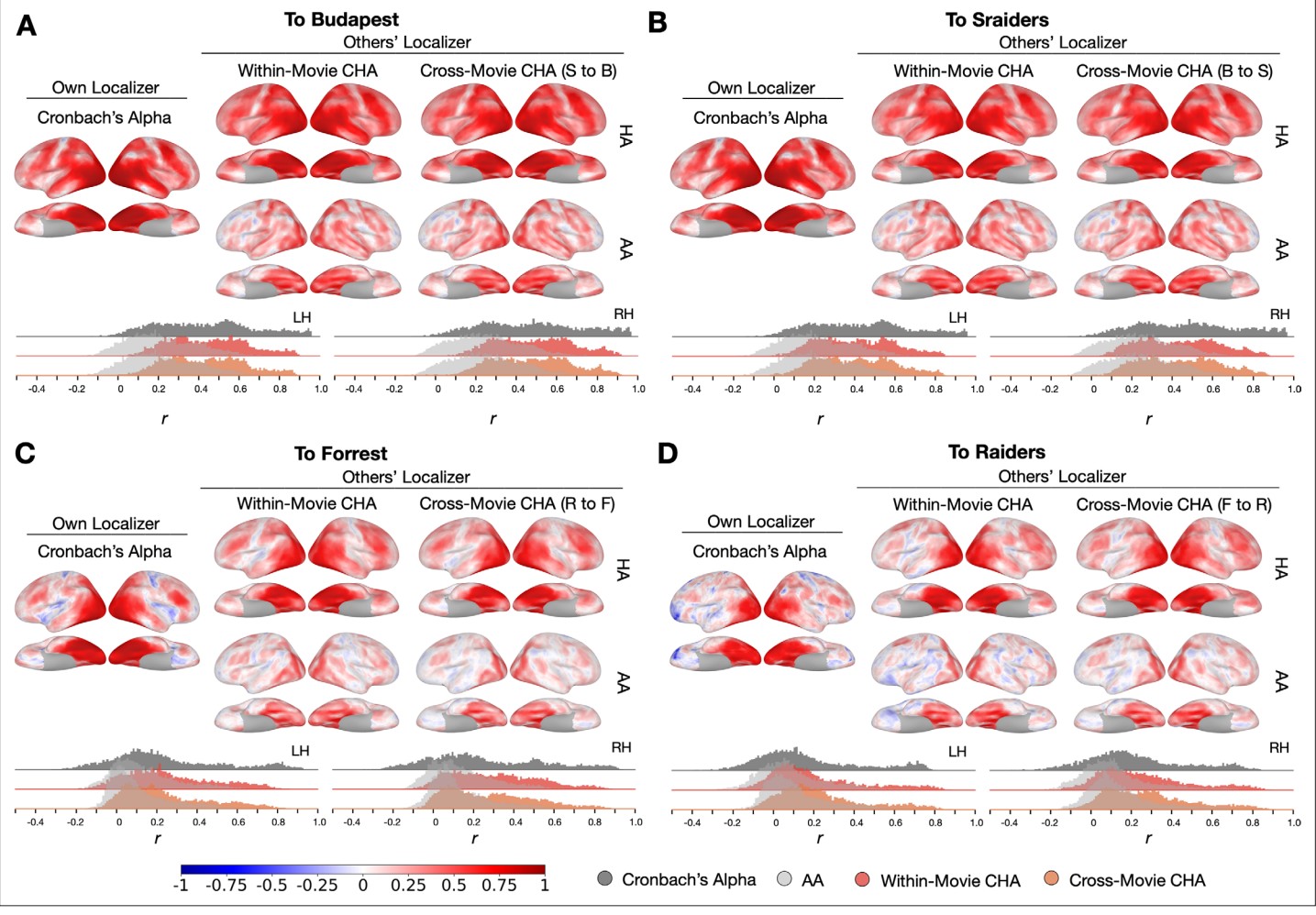

**Figure 4.** Searchlight analysis of Cronbach's alphas and prediction performances. (**A, B, C, and D**) The left-most column presents Cronbach's alphas of the own-localizer-based face-selective topographies in each dataset using a searchlight analysis (15 mm radius). The next two columns present local correlations (correlation maps) using the searchlight analysis between face-selective maps estimated from participants' own localizers and from other participants based on within-movie and between-movie connectivity hyperalignment (CHA) (hyperalignment [HA], top row) and surface alignment (AA, bottom row). Histogram plots present Cronbach's alphas (dark gray) and coefficients for the correlation maps above (estimated with CHA in color, with AA in light gray). The left and right hemisphere histograms were plotted separately. B to S: Budapest to Sraiders, S to B: Sraiders to Budapest, R to F: Raiders to Forrest, F to R: Forrest to Raiders.

The online version of this article includes the following figure supplement(s) for figure 4:

**Figure supplement 1.** Searchlight correlations.

**Figure supplement 2.** Distribution of correlation coefficients in major cortices.

**Figure supplement 3.** Searchlight analysis results for other categories.

**Figure supplement 4.** Advanced connectivity hyperalignment (CHA) improved prediction performances.

**Figure supplement 5.** Prediction performance changing with number of runs and participants.

**Figure supplement 6.** Predictions based on the 1-step and the 2-step methods.

**Figure supplement 7.** Comparing prediction performance between projecting time series and projecting contrast maps to the target participant.

**Figure supplement 8.** Prediction performance using participants with similar connectivity profiles.

**Figure supplement 9.** Similarities between fine-grained connectivities in two different movie-viewing tasks.

**Figure supplement 10.** Searchlight correlations with different searchlight sizes.

localizer used still images. For all categories, the dynamic localizer elicited stronger and broader category-selective activations than the static localizer, and the searchlight analysis showed that the dynamic localizer had higher reliabilities across the cortex, especially in regions that were selectively responsive to the target category. Due to differences between topographies activated by the dynamic and the static localizers, predictions across localizer types generated lower correlations than those within localizer types. For example, for the face-selective topographies, the dynamic localizer activated more areas than the static localizer (e.g., in superior temporal and frontal cortices). In the ventral temporal cortex, especially in the right hemisphere, both dynamic and static localizers performed well in the cross-localizer-type predictions. But in cortical areas where the static localizer did not match the dynamic localizer, predictions from the same dynamic localizer always outperformed the predictions from a different static localizer (*Figure 4—figure supplement 1*, *Figure 4—figure supplement 3*, and *Figure 4—figure supplement 10*). The low correlations were not because the prediction method failed but reflected the difference in the topographies activated by different types of localizers.

This study successfully illustrated that accurate individualized predictions are both robust and applicable across a variety of conditions, including movie types, languages, scanning parameters, and scanner models. Importantly, the intricate connectivity profiles remain consistent even when participants view entirely different movies, as evidenced by *Figure 4—figure supplement 9*, reinforcing the prediction's stability in various scenarios. However, all four datasets in this study only included typical participants with anatomically intact brains. An unanswered question is whether individualized topographies of neuropsychological populations with atypical cortical function (e.g., developmental prosopagnosis) or with lesioned brains (e.g., acquired prosopagnosis) could also be accurately predicted using the hyperalignment-based methods. Up to now, as far as we know, no previous literature has investigated this question. Beyond neuropsychological groups, it is also valuable to investigate how well the predictions will be across a wide range of age, from infants to the elderly. Future research is essential to adapt our algorithms to diverse populations.

In summary, our study demonstrated that accurate predictions of individualized category-selective topographies can be achieved with high fidelity using CHA across different naturalistic movie contents, across different scanners, and across different scanning parameters. Compared to traditional functional localizers, naturalistic stimuli are more ecologically valid, engaging multiple cognitive systems in parallel, and more friendly to participants. Our method not only can be applied directly to current public and in-lab datasets, but has the important potential to allow researchers to derive a broad range of topographies based on naturalistic movies and a normative database in the future. By building such a database that comprises various high-quality topographies and naturalistic stimuli, our study opens the gate to new research possibilities that could integrate high-level cognitive functions across datasets from laboratories worldwide.

## Materials and methods

### Datasets

#### The Budapest dataset

The *Budapest* dataset included 20 participants (mean age 27.2 years, 10 females) for this analysis. These participants were scanned while watching both *Grand Budapest Hotel* and *Raiders of the Lost Ark* and were a subset of the dataset in *Jiahui et al., 2020*. The *Grand Budapest Hotel* dataset contained five movie runs (~50 min, each part lasting 9–13 min each) and four dynamic localizer runs. Before entering the scanner, participants watched the first part of the movie (~45 min) outside. The rest of the movie was divided into five parts (each part lasting 9–13 min, ~50 min in total) and participants watched each part/run with audio. The dynamic localizer data were collected in a separate scanning section (*Pitcher et al., 2011*). This dataset comprised four blocked-designed runs (3.9 min each), and each run comprised 10 blocks (18 s each), two per category (faces, bodies, scenes, objects, and scrambled objects). Each block comprised six 3-s-long video clips in random order. Participants did a one-back task during the localizer scan to maintain attention.

All scans in the Grand Budapest Hotel dataset were acquired using a 3 T S Magnetom Prisma MRI scanner with a 32-channel head coil at the Dartmouth Brain Imaging Center. BOLD images were acquired in an interleaved fashion using gradient-echo echo-planar imaging with pre-scan normalization, fat suppression, multiband (i.e., simultaneous multi-slice) acceleration factor of 4 (using blipped

CAIPIRINHA), and no in-plane acceleration (i.e., GRAPPA acceleration factor of 1): TR/TE = 1000/33 ms, flip angle = 59°, resolution = 2.5 mm$^3$ isotropic voxels, matrix size = 96 × 96, FoV = 240 × 240 mm$^2$, 52 axial slices with full brain coverage and no gap, anterior-posterior phase encoding. See more details in *Visconti di Oleggio Castello et al., 2020*.

### The Sraiders dataset

The same participants were included for analysis in the *Sraiders* dataset as in the *Budapest* dataset. The movie *Raiders of the Lost Ark* was split into eight parts (~15 min each), and the first four parts were watched outside of the scanner prior to the scanning (~56 min). The later four parts were watched in the scanner (57 min) with audio (*Nastase, 2018*). The *Sraiders* dataset and the *Budapest* dataset shared the same dynamic localizer data. The *Sraiders* dataset was collected with the same scan protocols as the *Budapest* dataset (*Nastase, 2018*; *Feilong et al., 2022*).

### The Forrest dataset

This dataset contains scans from 15 adults (mean age 29.4 years, 6 females). Participants were scanned at the Otto-von-Guericke University in Germany and were native German speakers (*Hanke et al., 2016*; *Sengupta et al., 2016*). The dataset is publicly available at http://www.studyforrest.org/ (*Hanke et al., 2014*). A shortened version of the movie *Forrest Gump* was divided into eight parts with each part lasting approximately 15 min. Participants watched each part/run in the scanner with audio (*Hanke et al., 2016*). A category-selective localizer using still images was included in this dataset. This static localizer comprised four runs (5.2 min each). Each run comprised two 16 s blocks for each of the six categories (human faces, human bodies without heads, small objects, houses and outdoor scenes that include nature and street scenes, and phase scrambled images). In each block, 16 images from one category were displayed (900 ms display + 100 ms intertrial interval each). Participants were asked to do a one-back task to maintain attention.

Scanning was carried out using a whole-body 3 T Philips Achieva dStream MRI scanner equipped with a 32-channel head coil. Data were collected with gradient-echo, 2 s repetition time (TR), 30 ms echo time (TE), 90° flip angle, 1943 Hz/px bandwidth, and parallel acquisition with sensitivity encoding (SENSE) reduction factor 2. Each volume comprised 35 axial slices with anterior-to-posterior phase-encoding direction that were collected in ascending order, which mostly covered the entire brain. Each slice was 3.0 mm thick with a 10% inter-slice gap, and had a 240×240 mm$^2$ field-of-view comprising 80×80 3 mm$^2$ isotropic voxels. More acquisition parameters can be found in *Hanke et al., 2016*, and *Sengupta et al., 2016*.

### The Raiders dataset

A subset of nine participants from the original eleven participants (7 men, mean age = 24.8 years) participated in the face and object study at Dartmouth in *Haxby et al., 2011*, and were included in this dataset. The audio-visual movie *Raiders of the Lost Ark* was split into eight parts (~15 min each), similarly to those used in the *Sraiders* Dataset. Participants watched all eight parts in the scanner with audio (one part/per run). The *Raiders* dataset contains a static localizer that was similarly designed as in the *Forrest* dataset.

Brain images were acquired using a 3 T Philips Intera Achieva scanner with an eight-channel head coil at Dartmouth College. For the movie study, whole-brain volumes of 413-mm-thick sagittal images (TR = 2.5 s, TE = 35 ms, flip angle = 90°, 80×80 matrix, FOV = 240×240 mm$^2$, resolution = 0.938×0.938×1.0 mm$^3$) were obtained in an interleaved slice order. For more details see *Haxby et al., 2011*.

### MRI preprocessing

All datasets were preprocessed with fMRIPrep (*Esteban et al., 2019*), using version 20.1.1 for the *Budapest* dataset, 20.2.0 for the *Sraiders* dataset, 20.1.1 for the *Forrest* dataset, and 20.1.1 for the *Raiders* dataset. After fMRIPrep, functional data were projected onto a standard cortical surface aligned to the fsaverage template (*Fischl et al., 1999*) based on cortical folding patterns. The datasets were further preprocessed following *Jiahui et al., 2020*; *Feilong et al., 2018*. The datasets were resampled to a cortical mesh with 18,742 vertices across both hemispheres (approximately 3 mm

vertex spacing; 20,484 vertices before removing non-cortical vertices). Six motion parameters and their derivatives, global signal, framewise displacement (*Power et al., 2014*), six principal components from cerebrospinal fluid and white matter (*Behzadi et al., 2007*), and polynomial trends up to second order were rf out from both movie and localizer data for each run independently.

## Searchlight hyperalignment
### CHA (step 1)
Each participant's connectivity profile was built based on that participant's movie data. We first defined the connectivity seeds and targets. In this analysis, the connectivity seeds were the same as the surface cortical vertices. The connectivity targets were defined using a sparser cortical surface with 642 vertices in each hemisphere before removing the medial wall. We then centered a 13 mm searchlight on each of these vertices and computed the average time series for the searchlight over vertices from the denser cortical model. The mean time series was assigned to the center vertex to serve as the connectivity target. For each hemisphere, the connectivity profile was calculated as the correlation between the connectivity seeds in this hemisphere and the whole-brain 1175 connectivity targets. The connectivity profile of each participant was normalized to zero mean and unit variance for each connectivity seed before hyperalignment.

We used an optimized hyperalignment method that directly transforms one participant's connectivity profile to another participant's cortical space, without the interim step of projecting the connectome into a common model space (*Jiahui et al., 2020*). In detail, for each 15 mm searchlight, a participant's patterns of connectivity to targets were aligned to another participant's connectivity patterns using the Procrustes transformation. The transformation matrices from each searchlight in a hemisphere were then aggregated into a single transformation matrix for each pair of participants.

### Response hyperalignment
RHA was applied with the same steps as the CHA. The only difference is that instead of using connectivity profiles in each searchlight for each participant, we directly used the response pattern of the movie (time points of the movie × vertices in the searchlight) to align a pair of participants. In this method, response patterns in a pair of participants must be from neural responses to the same movie. Due to this restriction, RHA was only applied to participants from the same dataset.

### Advanced CHA
Using dense connectivity targets (e.g., using all 18,742 vertices on the surface) with anatomically aligned data usually generates poor functional correspondence across participants (*Busch et al., 2021*). It is, however, beneficial to include more targets for calculating connectivity patterns after the first iteration of CHA and repeated iterations to lead to a better solution by gradually aligning the information at finer scales.

We used six steps to further improve the CHA method. Step 1 was the initial CHA step as described above that was based on the raw anatomically aligned movie data. The resultant transformation matrices were applied to those movie runs, and the hyperaligned data were then used in step 2 to calculate new connectivity patterns and calculate new transformation matrices. We repeated this procedure iteratively six times and derived transformation matrices for each step. In steps 1, 2, and 3, 642×2 (icoorder3, before removing the medial wall) connectivity targets were defined with 13 mm searchlights. In steps 4 and 5, 2562×2 (icoorder 4, before removing the medial wall) connectivity targets were used with 7 mm searchlights to calculate target mean time series. In the final step 6, all 18,742 vertices were included as separate connectivity targets, using each vertex's time series rather than calculating the mean in a searchlight. Each step of this advanced CHA algorithm increased the prediction performance (*Figure 4—figure supplement 2*).

## Predicting individual contrast maps
### Estimating contrast maps from each participant's own localizer data
We estimated each participant's category-selective maps by calculating the unthresholded GLM univariate contrasts using his/her own localizer data in each run and averaging the t-values across all the localizer runs. We included face-, body-, scene-, and object-selective maps in the analysis. The

contrast maps in each category were calculated based on the contrast of the target category vs. all the other categories. For example, the face-selective map was calculated using faces vs. all the other categories in the localizer data (e.g., bodies, objects).

### Estimating contrast maps from other participants' localizer data

Transformation matrices from each participant to a target participant derived from hyperalignment were applied to the localizer runs of all other participants to project their localizer data into that target participant's cortical anatomy. These hyperaligned localizer runs and anatomical surface aligned localizer runs were used separately for GLM univariate analysis for each run in each other participant, and then averaged across the t-maps from all runs and all other participants to estimate the target participant's contrast maps for each category.

In summary, each participant's category-selective map was estimated based on that target participant's own localizer data and on all other participants' localizer data that was projected into that participant's cortical space using hyperalignment and anatomical surface alignment (see *Figure 1—figure supplement 1*). After obtaining these estimated maps, we calculated correlations between the target participant's category-selective maps based on his/her own localizer data and the maps estimated from other participants' data (hyperaligned or anatomically aligned). We also calculated Cronbach's alpha values (*Jiahui et al., 2020*; *Feilong et al., 2018*; *Jiahui et al., 2022*) across the multiple runs to measure the reliability of the category-selective maps for each participant and compared the correlations to the reliability values. Cronbach's alpha calculates the correlation score between localizer-based maps across the runs, and it reflects the amount of noise in maps based on individual localizer runs. Traditionally, the reliability was estimated based on split-half correlations. The common odd/even split measure underestimated reliability and necessitated recalculation of correlations between maps for only half the data to provide valid comparisons. In contrast, Cronbach's alpha involves all localizer runs and provides a more accurate statistical estimate of the reliability of the topographies estimated with localizer runs. To measure the local estimation performance and compare that to local reliabilities, we calculated correlations and Cronbach's alphas in searchlights with a radius of 15 mm.

## Acknowledgements

This work was supported by NSF grants 1607845 (JVH) and 1835200 (MIG), and NIH grant R01 MH127199 (JVH and MIG).

## Additional information

### Funding

| Funder | Grant reference number | Author |
| --- | --- | --- |
| National Science Foundation | 1607845 | James V Haxby |
| National Science Foundation | 1835200 | M Ida Gobbini |
| National Institute of Mental Health | MH127199 | James V Haxby |

The funders had no role in study design, data collection and interpretation, or the decision to submit the work for publication.

### Author contributions

Guo Jiahui, Conceptualization, Formal analysis, Investigation, Methodology, Software, Visualization, Writing – original draft, Writing – review and editing; Ma Feilong, Writing – original draft, Writing – review and editing, Investigation, Methodology; Samuel A Nastase, Writing – original draft, Methodology; James V Haxby, M Ida Gobbini, Conceptualization, Resources, Supervision, Funding acquisition, Investigation, Visualization, Methodology

## Author ORCIDs

Guo Jiahui (iD) https://orcid.org/0000-0002-1528-9025
James V Haxby (iD) http://orcid.org/0000-0002-6558-3118
M Ida Gobbini (iD) https://orcid.org/0000-0001-6727-7934

## Ethics

All participants gave their written informed consent to participate in the study. Data collection of the Forrest dataset was approved by the Ethics Committee of Otto-von-Guericke University (approval reference 37/13). Data collection of the other datasets (Raiders, Budapest, SRaiders) were approved by the Dartmouth Committee for the Protection of Human Subjects.

## Decision letter and Author response

Decision letter https://doi.org/10.7554/eLife.86037.sa1
Author response https://doi.org/10.7554/eLife.86037.sa2

---

# Additional files

## Supplementary files

• MDAR checklist

## Data availability

All data needed to evaluate the conclusions in the paper are present in the paper and/or the Supplementary Materials. Additional data and materials that support the findings of this study can be found at https://github.com/GUO-Jiahui/CHA_Cross-Movie_Prediction; (copy archived at *Jiahui, 2023*).

The following previously published datasets were used:

| Author(s) | Year | Dataset title | Dataset URL | Database and Identifier |
|---|---|---|---|---|
| Speck O, Hanke M, Baumgartner FJ, Ibe P, Kaule FR, Pollmann S, Speck O, Zinke W, Stadler J | 2018 | Forrest Gump | https://openneuro.org/datasets/ds000113 | OpenNeuro, ds000113 |
| Oleggio Castello MVD, Chauhan V, Jiahui G, Gobbini MI | 2020 | An fMRI dataset in response to 'The Grand Budapest Hotel', a socially-rich, naturalistic movie | https://openneuro.org/datasets/ds003017 | OpenNeuro, ds003017 |

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
