## [Editor Report]

This valuable study presents a tool for hyperaligning functional brain topography between individuals, which is based on fMRI connectivity data gathered when participants watched different movies. The tool is validated through strong correlations between functional topographic maps generated from a participant's own localizer data and those derived from other participants' data based on this hyperalignment, even when the training and target participants were drawn from different datasets. The study will potentially be of interest to researchers working with a wide range of fMRI datasets.

---

## [Decision Letter]

**Decision letter after peer review:**

Thank you for submitting your article "Cross-movie prediction of individualized functional topography" for consideration by *eLife*. Your article has been reviewed by 3 peer reviewers, including Ming Meng X as the Reviewing Editor and Reviewer #1, and the evaluation has been overseen by Chris Baker as the Senior Editor. The following individual involved in the review of your submission has agreed to reveal their identity: Zonglei Zhen (Reviewer #2).

Essential revisions:

1) Add discussion of the limit of the present hyper alignment approach: for example, to what extent the present hyper alignment approach would be applicable to individuals with atypical functional brain topography such as brain lesion patients with e.g., acquired prosopagnosia? Even in typical populations, while bilateral fusiform face areas can be identified in the majority through functional localizer scans, the left fusiform face area sometimes cannot be found. Moreover, many top-down factors are known to modulate functional brain topography. Due to these factors, brain responses and functional connectivity may be different even when the same subject watched the same movie twice (e.g., Cui et al., 2021).

2) Explain how the length of movie-viewing fMRI may affect the accuracy in predicting the idiosyncratic cortical topography? Similarly, how does the number of participants in the normative database affect the prediction of the category-selective topography? This information is important for the researchers who are interested in using the approach in their studies.

3) The data show that category-selective topography can be accurately estimated using connectivity hyper alignment, regardless of whether different movies are used to calculate the connectome and regardless of other data collection parameters. However, can the functional connectome from resting state fMRI accomplish the same as the movie-watching fMRI? If yes, this would expand the approach to much broader data.

4) The authors averaged the hyper-aligned functional localizer data from all of the subjects to predict individual category-selective topographies. As there is large spatial variability in the functional areas across subjects, averaging the data from many subjects may blur the boundaries of the functional areas. A better solution might be to average those subjects who show highly similar connectome to the target subjects.

5) Add discussion to clarify relations between the present hyperalignment approach and approaches in the literature that address the same question. Specifically, as reviewer #2 pointed out, 'Saygin and her colleagues have demonstrated that structural connectivity fingerprints can predict cortical selectivity for multiple visual categories across cortex (Osher DE et al., 2016, Cerebral Cortex; Saygin et al., 2011, Nat. Neurosci). I think there's a connection between those studies and the current study. If the author can discuss the connection between them, it may help us understand why CHA work so well.' And as reviewer #3 pointed out, 'the authors do not cite a paper that has already successfully demonstrated a functional alignment method that can address exactly this need: a connectivity-based Shared Response Model (cSRM; Nastase et al., 2020, NeuroImage). It would be relevant for the authors to consider the cSRM method in relation to their enhanced CHA method in detail. In particular, both the relative predictive performance as well as associated computational costs would be useful for researchers to understand in considering enhanced CHA for their applications.'

6) Justify the particular six step, iterative approach. That is: why were six steps chosen over any other number? At present, it is not clear if there is an explicit loss function that the authors are minimizing over their iterations. The relative computational cost of six iterations is also likely significant, particularly compared to previous hyperalignment algorithms. A more detailed theoretical understanding of why six iterations are necessary-or if other researchers could adopt a variable number according to the characteristics of their data-would significantly improve the transferability of this method.

7) The existing evaluations for enhanced CHA appear to be entirely based on image-derived correlations. That is, the authors compare the predicted image from CHA with the ground-truth image using correlation. While this provides promising initial evidence, correlation-based measures are often difficult to interpret given their sensitivity to image characteristics such as smoothness. Including Cronbach's α reliability as a baseline does not address this concern, as it is similarly an image-based statistic. It would be useful to see additional predictive experiments using frameworks such as time-segment classification, inter-subject decoding, or encoding models.

8) Make available the code for implementing CHA, or justify why this could not be done at the present.

*Reviewer #1 (Recommendations for the authors):*

In addition to adding more discussions on the limit of the present hyperalignment approach as I mentioned in the public review section, I would suggest more direct comparisons of the current CHA results and previous RHA results. I.e., perhaps consider moving Figure S2 to the main text?

*Reviewer #3 (Recommendations for the authors):*

– On L336 of The Raiders Dataset, the authors note that a subset of nine of the original eleven participants are included in the current experiments; however, from the current it is not obvious why two participants were excluded.

– Please confirm the radius of the searchlights used throughout the experiments. For example, in L361 of Connectivity Hyperalignment (Step One) the searchlight is described as 13mm radius, while on L370 of the same section it is a 15mm radius.

– In Figure S2, I noted the following two typos: In section (A), The second axis description should read "Values on the x-axis stand for correlations between each target participant's own localized-based topographies and topographies from other participants in the same dataset using CHA." In section (B), "Conbach's alphas" should be "Cronbach's alphas."

– In Figure S3, the in-figure legend (e.g., F to B) does not appear to relate to the figure content and is not explained in the figure description.

– It seems that code for implementing CHA is not currently available, as the GitHub repository listed in the Data Availability (but not in-text) does not contain executable code as far as I can tell. This would be particularly useful for other author's hoping to apply this method in their own datasets!

---

## [Author Response]

Essential revisions:1) Add discussion of the limit of the present hyper alignment approach: for example, to what extent the present hyper alignment approach would be applicable to individuals with atypical functional brain topography such as brain lesion patients with e.g., acquired prosopagnosia? Even in typical populations, while bilateral fusiform face areas can be identified in the majority through functional localizer scans, the left fusiform face area sometimes cannot be found. Moreover, many top-down factors are known to modulate functional brain topography. Due to these factors, brain responses and functional connectivity may be different even when the same subject watched the same movie twice (e.g., Cui et al., 2021).

We thank the reviewer for the suggestion and agree that it would be fascinating if the predictions can be made with high fidelity in neuropsychological populations. Although we are optimistic that our algorithm is able to generalize across diverse populations, to date, no previous literature has provided empirical evidence to illustrate the effectiveness, including optimizations and special applications beyond typical brains. Besides the neuropsychological population, it would also be valuable to study the generalization across a broad age range, for example, from infants to the elderly. The brain changes across age both anatomically and functionally, so it is a challenge to predict functional topographies based on a normative group that only includes young participants. With all these potential applications in mind, future research is needed to illustrate the efficacy, build the pipeline, and construct the representative normative groups to meet the requirements of accurate individualized predictions in diverse populations.

In typical populations, although participants have great individual variabilities in their functional topographies, for instance, some participants have distinguishable patches of activations in their left ventral temporal cortex while some participants don’t, our algorithms successfully captured these individualized differences in the prediction. Author response image 1 shows, as an example, the face-selective topographies of two individuals that have markedly different face-selective topographies on the left ventral temporal cortex. The left participant has prominent face-selective areas on the left ventral temporal cortex that are in similar sizes as the right side, while the right participant only has a few scattered small face-selective spots on the left side. No matter what their face-selective areas look like, our algorithm accurately recovered the individualized locations, shapes, and sizes, retaining the individual variability in the functional topographies.

**Author response image 1. sa2fig1:** 

Functional connectivity profiles based on naturalistic stimuli are very stable across the cortex, even when participants watch different movies. In Figure 4—figure supplement 9, the mean correlations of fine-scaled connectome for most searchlights (r = 15mm) when participants watched The Grand Budapest Hotel and the Raiders of the Lost Ark were generally around 0.8. The mean correlations were about 0.9 between the first and second half of the same movie although the stimuli contents were different between the two halves. Thus, the fine-grained functional connectivity profiles remain highly stable and reliable across movie contents, which contributes to the robustness of cross-movie, time, and other parameters (e.g., scanner models, scanning parameter) predictions using our algorithms.We added a paragraph in the discuss section to address the concerns (page 18-19):

“This study successfully illustrated that accurate individualized predictions are both robust and applicable across a variety of conditions, including movie types, languages, scanning parameters, and scanner models. Importantly, the intricate connectivity profiles remain consistent even when participants view entirely different movies, as evidenced by Figure 4—figure supplement 9, reinforcing the prediction's stability in various scenarios. However, all four datasets in this study only included typical participants with anatomically intact brains. An unanswered question is whether individualized topographies of neuropsychological populations with atypical cortical function (e.g., developmental prosopagnosics) or with lesioned brains (e.g., acquired prosopagnosics) could also be accurately predicted using the hyperalignment-based methods. Up to now, as far as we know, no previous literature has investigated this question. Beyond neuropsychological groups, it is also valuable to investigate how well the predictions will be across a wide range of age, from infants to the elderly. Future research is essential to adapt our algorithms to diverse populations.”

2) Explain how the length of movie-viewing fMRI may affect the accuracy in predicting the idiosyncratic cortical topography? Similarly, how does the number of participants in the normative database affect the prediction of the category-selective topography? This information is important for the researchers who are interested in using the approach in their studies.

To investigate the influence of movie-viewing data length and the number of participants in the normative database on prediction performance, we systematically varied these parameters. Specifically, we altered the number of runs utilized in the analysis for both the normative and target data and experimented with varying the number of participants in the normative dataset using the Budapest and the Sraiders datasets. We have included a new Figure 4—figure supplement 5 to present a summary of these findings.

The results reveal that both within-dataset and between-dataset prediction performances are positively correlated with the length of movie-viewing fMRI data used for both the normative and target groups. A similar trend was observed with respect to the number of participants included in the normative dataset. It is important to highlight, though, that, even when analyzing as little as one run of movie-viewing data—roughly 10-15 minutes, our hyperalignment-based prediction performance was significantly higher than that achieved using traditional surface alignment. This held true even when the normative dataset included as few as five participants.

In summary, our results show that prediction performance generally improves with longer movie-viewing sessions and larger normative datasets. However, it is noteworthy that even with minimal data—10 minutes of movie-viewing and a small number of participants in the normative dataset—our algorithm still outperforms traditional surface alignment methods significantly.

We also added sentences in the Discussion section (page 15):

“We investigated the influence of naturalistic movie length and the size of the training group on the prediction accuracy of individualized functional topographies. By incrementally increasing both the number of movie runs in the training and target dataset and the participants in the training group in the Budapest and Sraiders dataset, we observed enhanced prediction accuracy (Figure 4—figure supplement 5). Notably, even with just one movie run in the training or target dataset, or with a mere five participants in the training group, our prediction performance (Pearson *r*) ranged from about 0.6 to 0.7. This accuracy significantly outperformed results obtained using surface-based alignment.”

3) The data show that category-selective topography can be accurately estimated using connectivity hyper alignment, regardless of whether different movies are used to calculate the connectome and regardless of other data collection parameters. However, can the functional connectome from resting state fMRI accomplish the same as the movie-watching fMRI? If yes, this would expand the approach to much broader data.

We agree with the reviewer that demonstrating the applicability of the resting state data will expand the application scenarios of this approach. Most previous findings on resting state connectivity, including the comparison between the naturalistic and the resting state paradigms, focused on the macro-scale similarities and differences (e.g., Samara et al., 2023). Very few studies have investigated the fine-scaled connectome based on resting state data. The study on connectivity hyperalignment (Guntupalli et al., 2018) demonstrated a shared fine-scale connectivity structure among individuals that co-exists with the common coarse-scale structure and built the algorithm to successfully hyperalign individuals to the shared fine-scaled space. Another study from our lab (Feilong et al., 2021) revealed that the fine-scaled connectivity profiles in both resting and task states are highly predictive of general intelligence, indicating reliable and biologically relevant fine-scaled resting state connectome structures. Thus, it is highly plausible that our approach is able to be generalized to the resting state data, generating significantly better predictions of individualized functional topographies than traditional surface alignment. However, due to the limitations of the current datasets, we do not have resting state data available in the current datasets to perform this analysis. We are in the process of collecting new data to explore this hypothesis in future work.

We added sentences to the Discussion section to discuss this idea (page 18):

“Studies comparing movie-viewing and resting state functional connectivity have shown that both paradigms yield overlapping macroscale cortical organizations (*29*), though naturalistic viewing introduces unique modality-specific hierarchical gradients. However, there remains a gap in research comparing the fine-scaled connectomes of naturalistic and resting state paradigms. Guntupalli and colleagues (*14*) revealed a shared fine-scale structure that coexists with the coarse-scale structure, and connectivity hyperalignment successfully improved intersubject correlations across a wide variety of tasks. Feilong et al. (*13*) noted that the fine-scaled connectivity profiles in both resting and task states are highly predictive of general intelligence. This suggests a reliable and biologically relevant fine-scale resting state connectivity structure among individuals. Therefore, it is plausible that individualized functional topography could be effectively estimated using resting state functional connectivity, expanding the applicability of our approach. Future studies are needed to explore this direction.”

4) The authors averaged the hyper-aligned functional localizer data from all of the subjects to predict individual category-selective topographies. As there is large spatial variability in the functional areas across subjects, averaging the data from many subjects may blur the boundaries of the functional areas. A better solution might be to average those subjects who show highly similar connectome to the target subjects.

We appreciate the reviewer’s insightful question about optimizing prediction performance by selecting participants most similar in functional connectivity to the target individuals. This is a promising direction and difficult problem as well. Our approach is based on fine-scale connectome to hyperalign participants, thus different groups of participants may be similar to the target participant in different searchlights. In addition, based on results discussed in the response to Q2, the more participants included in the normative dataset, the better the prediction performance. Thus, there is a trade-off between the number of participants included in the normative dataset for the prediction and the overall similarity of those participants to the target participant.

To quantitatively explore this idea, we used a searchlight in the right ventral temporal cortex, roughly at the location of posterior fusiform face area (pFFA). We sorted participants by their connectome similarity to each target participant and then examined prediction performance based on either the top nine most similar participants or the bottom nine least similar participants. Our results, presented in Figure 4—figure supplement 8, reveal that hyperalignment consistently outperforms surface alignment regardless of the subset of participants used. Notably, using the nine most similar participants did not significantly alter prediction performance (Tukey Test, z = -0.09, p = 0.996), while using the least similar participants did negatively impact it (Tukey Test, z = 2.492, p = 0.034). Interestingly, the stability of hyperalignment-based predictions remained high even when only a subset of participants was used, contrasting with the variability observed in surface-alignment-based predictions.

Overall, these findings suggest that while selecting functionally similar participants is a promising avenue for future optimization, the process will require nuanced, searchlight-specific criteria. Each searchlight may necessitate its own set of optimal participants to balance between the performance boost from having more participants and the fidelity gained from participant similarity.

We added the following to the discussion in the manuscript (page 16):

“In our study, we used fine-scale connectomes, noting that some participants are more similar to the target participant in specific searchlights. It is an interesting question whether predictions could be enhanced by exclusively selecting those more similar participants for the target participant. To explore this option, we examined a searchlight in the right ventral temporal cortex that was roughly at the location of the posterior fusiform area (pFFA) using the top and bottom nine participants similar to each target participant measured by their fine-scale connectome similarities in the budapest dataset. Generally, using all or part of the participants for the prediction generated similar results (Figure 4—figure supplement 8). Compared to using all the participants, using only the top nine participants who are the most similar to the target participants did not significantly improve the prediction (Tukey Test, z = -0.09, p = 0.996), but using only the bottom nine participants generated significantly lower prediction accuracies (Tukey Test, z = 2.492, p = 0.034). This suggests a trade-off between the number of participants included in the prediction and the similarity of the participants. Future studies are needed to explore the optimal threshold for the number of participants included for each searchlight to refine the algorithm.”

5) Add discussion to clarify relations between the present hyperalignment approach and approaches in the literature that address the same question. Specifically, as reviewer #2 pointed out, 'Saygin and her colleagues have demonstrated that structural connectivity fingerprints can predict cortical selectivity for multiple visual categories across cortex (Osher DE et al., 2016, Cerebral Cortex; Saygin et al., 2011, Nat. Neurosci). I think there's a connection between those studies and the current study. If the author can discuss the connection between them, it may help us understand why CHA work so well.' And as reviewer #3 pointed out, 'the authors do not cite a paper that has already successfully demonstrated a functional alignment method that can address exactly this need: a connectivity-based Shared Response Model (cSRM; Nastase et al., 2020, NeuroImage). It would be relevant for the authors to consider the cSRM method in relation to their enhanced CHA method in detail. In particular, both the relative predictive performance as well as associated computational costs would be useful for researchers to understand in considering enhanced CHA for their applications.'

We thank the reviewer for raising this point that provides us with the chance of clarifying how our approach differs with methods previously reported in the literature. The computational logic underlying our approach is that we derived the transformation matrices between the training and the target participants in the high-dimensional space based on functional connectivity calculated from the movie data. Then, we applied these transformation matrices to the training participant’s localizer data to accomplish the prediction. On the other hand, Saygin and colleagues directly used diffusion-weighted imaging (DWI) data and predicted participants’ functional responses based on the anatomical-functional correspondence. They evaluated the prediction by calculating the mean absolute errors (AE) of the difference between the actual and predicted contrast responses. Although AE linearly increases with the quality of the prediction, it is difficult to measure the prediction performance of the shape, size, and location of the functional areas precisely using this mean value. With our algorithm, we were able to predict the general location and size of the areas and recover the individualized shapes, generating more powerful predictions. We also used the searchlight analysis to evaluate the performance across the cortex systematically. In addition, Osher et al. (2016) and Saygin et al. (2012) always have a few participants failing to show better predictions based on the connectivity than the group averaged method. Our algorithm is more stable, as all participants across all four datasets had better predicted performance using our algorithm than using the group average. However, although we did not directly use the anatomical-functional correspondence with DWI, the relationships between individual structural connectivity and cortical visual category selectivity could be one of the biological underpinnings that contribute to this robust and accurate prediction.

The Connectivity-Based Shared Response Model (cSRM, Nastase et al., 2020) offers an alternative framework for aligning individuals through functional connectivity. While the overarching aim of cSRM and our methodology converges, substantial differences emerge in the respective implementation and application between the two methods that make our approach the more suitable for predicting individualized topographies. The most significant difference between the two is that, instead of focusing on within-individual connectivity profiles, cSRM used inter-subject functional connectivity (ISFC) in the initial step. This design requires that all participants must have time-locked time series, making the algorithm unusable for cross-content prediction and making it incompatible with resting-state data. Our approach, on the other hand, does not require time-locked stimuli, thereby offering a more flexible framework that permits generalization across different types of stimuli and experimental settings and enables bringing data across laboratories across the world together. Secondly, cSRM predominantly focuses on Region of Interest (ROI) analyses, whereas our model employs searchlight-based analyses designed to comprehensively cover the entire cortical sheet. Whole-brain coverage is needed to generate the topography that reflects the patterns across the cortex. Finally, with the optimized 1step method, our approach directly hyeraligns the training and target participants together, avoiding the accumulation of errors from the intermediate common space. cSRM, with an implementation similar to the classic connectivity hyperalignment, creates and hyperaligns all participants to a shared information space. In summary, while our approach and cSRM share a similar theoretical foundation, our approach has been specifically optimized to address the challenges and complexities in predicting individualized whole-brain functional topographies. Moreover, our approach demonstrates a remarkable ability to generalize across a variety of contexts and stimuli, offering a significant advantage in dealing with diverse experimental settings and datasets.

We have added the contents to the Discussion section (page 16-17):

“By leveraging transformation matrices obtained from hyperaligning participants based on movie-viewing data, we successfully mapped these relationships to the training participants’ localizer data, enabling robust predictions. Prior work employing diffusion-weighted imaging (DWI) has underscored the link between anatomical connectivity and category selectivity across diverse visual fields (*22, 23*) and has established a notable congruence between structural and functional connectivities (*24*). These findings suggest that the unique anatomical connectivity patterns of individuals may serve as a foundational mechanism, contributing to the stable finescale functional connectome that underpins our approach. The connectivity-based Shared Response Model (cSRM) proposed by Nastase and colleagues (*25*) used connectivity to functionally align individuals similar to the connectivity hyperalignment algorithm. While both approaches share overarching goals, they diverge considerably in implementation and application. First and most important, cSRM used inter-subject functional connectivity (ISFC) rather than within-subject functional connectivity to initially estimate the connectome. As a result, cSRM requires participants to have time-locked fMRI time series. Therefore, unlike our algorithm, the cSRM approach does not support cross-content applications and also is not suitable for use with resting-state data. Second, cSRM is implemented based on a predefined cortical parcellation rather than the overlapping, regularly-spaced cortical searchlights applied in our method which are not constrained by areal borders. For the application, cSRM has mainly been used to do ROI analysis rather than the estimation of the whole-brain topography that requires broader coverage of the cortex with a searchlight analysis. Third, our method is specifically designed to work in each individual’s space, while cSRM decomposes data across subjects into shared and subject specific transformations, focusing on a communal connectivity space. In summary, although cSRM presents a promising alternative for similar aims, its current implementation precludes it from fulfilling the range of applications for which our method is optimized.”

6) Justify the particular six step, iterative approach. That is: why were six steps chosen over any other number? At present, it is not clear if there is an explicit loss function that the authors are minimizing over their iterations. The relative computational cost of six iterations is also likely significant, particularly compared to previous hyperalignment algorithms. A more detailed theoretical understanding of why six iterations are necessary-or if other researchers could adopt a variable number according to the characteristics of their data-would significantly improve the transferability of this method.

In the advanced connectivity hyperalignment implementation, we gradually increased the number of targets. The six steps were not intentionally chosen but were the result of the increase to the maximum number of fine-grained targets, namely single cortical vertices.

Our datasets were resampled to the cortical mesh with 18,742 vertices across both hemispheres (approximately 3 mm vertex spacing; icoorder 5; 20,484 vertices before removing non-cortical vertices). Step 1 was the classic standard connectivity hyperalignment implementation based on the anatomically-aligned data. Since using dense connectivity targets (e.g., using all 18742 vertices on the surface) with anatomically-aligned data generates poor functional correspondence across participants (Busch et al., 2021), we used 1,284 vertices (icoorder 3, before removing the medial wall) as connectivity targets in step 1. However, it is beneficial to include more targets for calculating connectivity patterns after the first iteration of connectivity hyperalignment and repeated iterations to lead to a better solution by gradually aligning the information at finer scales. To better align across participants, we iterated the alignment for another two times (step 2 and step 3) with the same number of 1,284 coarse connectivity targets to ensure improved alignment before increasing the number of targets in the later steps. In step 4, we increased the number of targets to 5,124 (icoorder 4, before removing the medial wall), and iterated with this number of vertices for two times in total (step 4 and step 5) before using all vertices as targets. In the final step (step 6), all vertices were used as connectivity targets.

It is true that the multiple iteration steps largely increased the computational complexity compared to the classic connectivity hyperalignment, but the prediction increase was steady across all datasets and became comparable to response hyperalignment performance which requires time-locked stimuli. We did not use an explicit loss function in the algorithm, but followed the natural progression of the number of potential connectivity targets in the implementation. On the other hand, the difference between the performance of the improved and the classic connectivity hyperalignment was relatively small (difference of *r* < 0.05), which indicates the effectiveness of our classic algorithm. It is up to the researchers’ own options to adopt the number of iterations and the pace of increasing the number of targets in each step. If computational resources are limited or if a shorter total computational time is the primary priority, using the classic connectivity hyperalignment may be the best option to balance the trade-offs.

The Materials and methods section had the details of the implementation (page 22-23):

“Using dense connectivity targets (e.g., using all 18742 vertices on the surface) with anatomically-aligned data usually generates poor functional correspondence across participants (33). It is, however, beneficial to include more targets for calculating connectivity patterns after the first iteration of connectivity hyperalignment and repeated iterations to lead to a better solution by gradually aligning the information at finer scales.

We used six steps to further improve the connectivity hyperalignment method. Step 1 was the initial connectivity hyperalignment step as described above that was based on the raw anatomically aligned movie data. The resultant transformation matrices were applied to those movie runs, and the hyperaligned data were then used in step 2 to calculate new connectivity patterns and calculate new transformation matrices. We repeated this procedure iteratively six times and derived transformation matrices for each step. In steps 1, 2, and 3, 642 × 2 (icoorder3, before removing the medial wall) connectivity targets were defined with 13 mm searchlights. In step 4 and 5, 2562 × 2 (icoorder 4, before removing the medial wall) connectivity targets were used with 7 mm searchlights to calculate target mean time series. In the final step 6, all 18742 vertices were included as separate connectivity targets, using each vertex’s time series rather than calculating the mean in a searchlight. Each step of this advanced connectivity hyperalignment algorithm increased the prediction performance (Figure 4—figure supplement 2).”

But to help the readers understand the logic of the advanced connectivity hyperalignment algorithm used in this study, we expanded the Discussion section (page 15):

“Because using dense connectivity targets (e.g., using all vertices as connectivity targets) with anatomically-alignment data often leads to suboptimal alignment across participants (*33*), we started with coarse connectivity targets and gradually increased the number of connectivity targets to form a denser representation of connectivity profiles. The iterations improved the prediction performance step by step, and at the final step (step 6, all vertices were used as connectivity targets) in this analysis, the enhanced CHA generated comparable performance with RHA (Figure 4—figure supplement 4).”

7) The existing evaluations for enhanced CHA appear to be entirely based on image-derived correlations. That is, the authors compare the predicted image from CHA with the ground-truth image using correlation. While this provides promising initial evidence, correlation-based measures are often difficult to interpret given their sensitivity to image characteristics such as smoothness. Including Cronbach's α reliability as a baseline does not address this concern, as it is similarly an image-based statistic. It would be useful to see additional predictive experiments using frameworks such as time-segment classification, inter-subject decoding, or encoding models.

We appreciate the reviewer’s concern regarding the stability of local correlations in relation to image characteristics. To address this, we conducted additional analysis using different searchlight sizes (with radii of 10 mm, 15 mm, and 20 mm) to evaluate the predicted categoryselective maps, focusing specifically on the Budapest dataset. The local correlations between the predicted category-selective maps (obtained using enhanced CHA) and participants’ own maps based on classic localizer runs were calculated for each searchlight. We averaged these correlations across participants and plotted the resulting maps, as shown in Figure 4—figure supplement 10. Although using a larger searchlight radius is similar to employing a larger smoothing kernel, the results remained relatively stable across different searchlight sizes, particularly in regions selectively responsive to the specific category. This stability suggests that while the evaluation may be influenced by image-related features, the conclusion would remain consistent under varying parameters.

As for the use of enhanced CHA, it serves as an optimized version of the classic CHA, specifically designed for predicting individualized functional topographies. Evaluating prediction performance in our study is based on t-value contrast maps for each participant. Given this, it's unclear how time-segment classification or other decoding/encoding models could be appropriately implemented for performance evaluation. However, prior research from our lab has already established the effectiveness of classic CHA. Specifically, Guntupalli et al. (2018) showed that classic CHA significantly improved intersubject correlations (ISC) of connectivity profiles across the cortex. They also revealed that CHA captured fine-scale variations in connectivity profiles for nearby cortical nodes across participants and led to improved betweensubject multivariate pattern classification accuracies (bsMVPC) of movie segments. These findings serve as robust evidence for the effectiveness of classic CHA, laying the groundwork for our enhanced CHA approach.

We added Figure 4—figure supplement 10 to the supplementary material:

8) Make available the code for implementing CHA, or justify why this could not be done at the present.

We will make the implementation code available once the article is accepted.

Reviewer #1 (Recommendations for the authors):In addition to adding more discussions on the limit of the present hyperalignment approach as I mentioned in the public review section, I would suggest more direct comparisons of the current CHA results and previous RHA results. I.e., perhaps consider moving Figure S2 to the main text?

We thank the reviewer for pointing this out. To help readers better understand the comparison between CHA and RHA results, we moved Figure S2A (current Figure 2—figure supplement 1A) to the main text and combined it with Figure 1 as panel D.

The corresponding content in the manuscript is in the Results section (page 4):

“Estimates using CHA to calculate transformation matrices were also equivalent to estimates using RHA (Figure 1D). RHA, however, requires that all subjects watch the same movie, whereas CHA can use connectivity matrices derived from responses to different movies, potentially making our new approach more flexible.”

Reviewer #3 (Recommendations for the authors):– On L336 of The Raiders Dataset, the authors note that a subset of nine of the original eleven participants are included in the current experiments; however, from the current it is not obvious why two participants were excluded.

Nine of the total eleven original participants have the localizer data, thus, only these nine participants were included in this study.

– Please confirm the radius of the searchlights used throughout the experiments. For example, in L361 of Connectivity Hyperalignment (Step One) the searchlight is described as 13mm radius, while on L370 of the same section it is a 15mm radius.

The details are correct. We used slightly smaller sized (13 mm) searchlights when building the connectivity profiles and the common sized (15 mm) searchlights in the functional alignment.

– In Figure S2, I noted the following two typos: In section (A), The second axis description should read "Values on the x-axis stand for correlations between each target participant's own localized-based topographies and topographies from other participants in the same dataset using CHA." In section (B), "Conbach's alphas" should be "Cronbach's alphas."

We made the revision as suggested for section (B). We moved Figure S2A (current Figure 2figure supplement 1A) to current Figure 1D and kept the figure caption to explicitly describe that the predicted topographies were estimated from other participants in the same dataset.

– In Figure S3, the in-figure legend (e.g., F to B) does not appear to relate to the figure content and is not explained in the figure description.

For each dataset, we included RHA, within-movie CHA, AA, and all possible cross-movie CHA predictions. So for each individual participant in the figure, all colored dots reflecting the listed contents were plotted. We apologize for the confusion, and added the explanation of the legends in the figure caption.

– It seems that code for implementing CHA is not currently available, as the GitHub repository listed in the Data Availability (but not in-text) does not contain executable code as far as I can tell. This would be particularly useful for other author's hoping to apply this method in their own datasets!

We will make the implementation code available once the article is accepted.

References

Feilong, M., Guntupalli, J. S., and Haxby, J. V. (2021). The neural basis of intelligence in finegrained cortical topographies. *eLife*, *10*, e64058. https://doi.org/10.7554/*eLife*.64058

Guntupalli, J. S., Feilong, M., and Haxby, J. V. (2018). A computational model of shared finescale structure in the human connectome. *PLOS Computational Biology*, *14*(4), e1006120. https://doi.org/10.1371/journal.pcbi.1006120

Guntupalli, J. S., Hanke, M., Halchenko, Y. O., Connolly, A. C., Ramadge, P. J., and Haxby, J. V. (2016). A Model of Representational Spaces in Human Cortex. Cerebral Cortex, 26(6), 2919–2934. https://doi.org/10.1093/cercor/bhw068

Jiahui, G., Feilong, M., Visconti di Oleggio Castello, M., Guntupalli, J. S., Chauhan, V., Haxby, J. V., and Gobbini, M. I. (2020). Predicting individual face-selective topography using naturalistic stimuli. *NeuroImage*, *216*, 116458. https://doi.org/10.1016/j.neuroimage.2019.116458

Jiahui, G., Feilong, M., Visconti di Oleggio Castello, M., Nastase, S. A., Haxby, J. V., and Gobbini, M. I. (2023). Modeling naturalistic face processing in humans with deep convolutional neural networks. *Proceedings of the National Academy of Sciences*, *120*(43), e2304085120. https://doi.org/10.1073/pnas.2304085120

Nastase, S. A., Liu, Y.-F., Hillman, H., Norman, K. A., and Hasson, U. (2020). Leveraging shared connectivity to aggregate heterogeneous datasets into a common response space. *NeuroImage*, *217*, 116865. https://doi.org/10.1016/j.neuroimage.2020.116865

Osher, D. E., Saxe, R. R., Koldewyn, K., Gabrieli, J. D. E., Kanwisher, N., and Saygin, Z. M. (2016). Structural Connectivity Fingerprints Predict Cortical Selectivity for Multiple Visual Categories across Cortex. Cerebral Cortex (New York, NY), 26(4), 1668–1683. https://doi.org/10.1093/cercor/bhu303

Samara, A., Eilbott, J., Margulies, D. S., Xu, T., and Vanderwal, T. (2023). Cortical gradients during naturalistic processing are hierarchical and modality-specific. *NeuroImage*, *271*, 120023. https://doi.org/10.1016/j.neuroimage.2023.120023

Saygin, Z. M., Osher, D. E., Koldewyn, K., Reynolds, G., Gabrieli, J. D. E., and Saxe, R. R. (2012). Anatomical connectivity patterns predict face selectivity in the fusiform gyrus. Nature Neuroscience, 15(2), 321–327. https://doi.org/10.1038/nn.3001